# Asynchronous Actor-Critic for Multi-Agent Reinforcement Learning

**Yuchen Xiao**
Khoury College of Computer Sciences
Northeastern University
Boston, MA 02115
xiao.yuch@northeastern.edu

**Weihao Tan**
Khoury College of Computer Sciences
Northeastern University
Boston, MA 02115
w.tan@northeastern.edu

**Christopher Amato**
Khoury College of Computer Sciences
Northeastern University
Boston, MA 02115
c.amato@northeastern.edu

## Abstract

Synchronizing decisions across multiple agents in realistic settings is problematic since it requires agents to wait for other agents to terminate and communicate about termination reliably. Ideally, agents should learn and execute asynchronously instead. Such asynchronous methods also allow temporally extended actions that can take different amounts of time based on the situation and action executed. Unfortunately, current policy gradient methods are not applicable in asynchronous settings, as they assume that agents synchronously reason about action selection at every time step. To allow asynchronous learning and decision-making, we formulate a set of asynchronous multi-agent actor-critic methods that allow agents to directly optimize asynchronous policies in three standard training paradigms: decentralized learning, centralized learning, and centralized training for decentralized execution. Empirical results (in simulation and hardware) in a variety of realistic domains demonstrate the superiority of our approaches in large multi-agent problems and validate the effectiveness of our algorithms for learning high-quality and asynchronous solutions.

## 1 Introduction

In recent years, multi-agent policy gradient methods using the actor-critic framework have achieved impressive success in solving a variety of cooperative and competitive domains [Baker et al., 2020, Du et al., 2019, Foerster et al., 2018, Du et al., 2021, Iqbal and Sha, 2019, Li et al., 2019, Lowe et al., 2017, Su et al., 2021, Vinyals et al., 2019, Wang et al., 2020a, 2021a, Yang et al., 2020a, Zhou et al., 2020]. However, as these methods assume synchronized primitive-action execution over agents, they struggle to solve large-scale real-world multi-agent problems that involve long-term reasoning and asynchronous behavior.

Temporally-extended actions have been widely used in both learning and planning to improve scalability and reduce complexity. For example, they have come in the form of motion primitives [Dalal et al., 2021, Stulp and Schaal, 2011], skills [Konidaris et al., 2011, 2018], spatial action maps [Wu et al., 2020] or macro-actions [He et al., 2010, Hsiao et al., 2010, Lee et al., 2021, Theocharous and Kaelbling, 2004]. The idea of temporally-extended actions has also been incorporated into multi-agent approaches. In particular, we consider the *Macro-Action Decentralized Partially Observable*

*Markov Decision Process* (MacDec-POMDP) [Amato et al., 2014, 2019]. The MacDec-POMDP is a general model for cooperative multi-agent problems with partial observability and (potentially) different action durations. As a result, agents can start and end macro-actions at different time steps so decision-making can be asynchronous.

The MacDec-POMDP framework has shown strong scalability with planning-based methods (where the model is given) [Amato et al., 2015a,b, Hoang et al., 2018, Omidshafiei et al., 2016, 2017a]. In terms of multi-agent reinforcement learning (MARL), there have been many hierarchical approaches, they don't typically address asynchronicity since they assume agents' have high-level decisions with the same duration [de Witt et al., 2019, Han et al., 2019, Nachum et al., 2019, Wang et al., 2020b, 2021b, Xu et al., 2021, Yang et al., 2020b]. Only limited studies have considered asynchronicity [Chakravorty et al., 2019, Menda et al., 2019, Xiao et al., 2019], yet, none of them provides a general formulation for multi-agent policy gradients that allows agents to asynchronously learn and execute.

In this paper, we assume a set of macro-actions has been predefined for each domain. This is well-motivated by the fact that, in real-world multi-robot systems, each robot is already equipped with certain controllers (e.g., a navigation controller, and a manipulation controller) that can be modeled as macro-actions [Amato et al., 2015a, Omidshafiei et al., 2017a, Wu et al., 2021a, Xiao et al., 2019]. Similarly, as it is common to assume primitive actions are given in a typical RL domain, we assume the macro-actions are given in our case. The focus of the policy gradient methods is then on learning high-level policies over macro-actions.[1]

Our contributions include a set of macro-action-based multi-agent actor-critic methods that generalize their primitive-action counterparts. First, we formulate a *macro-action-based independent actor-critic* (Mac-IAC) method. Although independent learning suffers from a theoretical curse of environmental non-stationarity, it allows fully online learning and may still work well in certain domains. Second, we introduce a *macro-action-based centralized actor-critic* (Mac-CAC) method, for the case where full communication is available during execution. We also formulate a centralized training for decentralized execution (CTDE) paradigm [Kraemer and Banerjee, 2016, Oliehoek et al., 2008] variant of our method. CTDE has gained popularity since such methods can learn better decentralized policies by using centralized information during training. Current primitive-action-based multi-agent actor-critic methods typically use a centralized critic to optimize each decentralized actor. However, the asynchronous joint macro-action execution from the centralized perspective could be very different with the completion time being very different from each agent's decentralized perspective. To this end, we first present a *Naive Independent Actor with Centralized Critic* (Naive IACC) method that naively uses a joint macro-action-value function as the critic for each actor's policy gradient estimation; and then propose a novel *Independent Actor with Individual Centralized Critic* (Mac-IAICC) method that learns individual critics using centralized information to address the above challenge.

We evaluate our proposed methods on diverse macro-action-based multi-agent problems: a benchmark Box Pushing domain [Xiao et al., 2019], a variant of the Overcooked domain [Wu et al., 2021b] and a larger warehouse service domain [Xiao et al., 2019]. Experimental results show that our methods are able to learn high-quality solutions while primitive-action-based methods cannot, and show the strength of Mac-IAICC for learning decentralized policies over Naive IAICC and Mac-IAC. Decentralized policies learned by using Mac-IAICC are successfully deployed on real robots to solve a warehouse tool delivery task in an efficient way. To our knowledge, this is the first general formalization of macro-action-based multi-agent actor-critic frameworks for the three state-of-the-art multi-agent training paradigms.

## 2 Background

### 2.1 MacDec-POMDPs

The macro-action decentralized partially observable Markov decision process (MacDec-POMDP) [Amato et al., 2014, 2019] incorporates the *option* framework [Sutton et al., 1999] into the Dec-POMDP by defining a set of macro-actions for each agent. Formally, a MacDec-POMDP is defined by a tuple $\langle I, S, A, M, \Omega, \zeta, T, R, O, Z, \mathbb{H}, \gamma \rangle$, where $I$ is a set of agents; $S$ is the environ-

---

[1]Our approach could potentially also be applied to other models with temporally-extended actions [Omidshafiei et al., 2017a].

mental state space; $A = \times_{i \in I} A_i$ is the joint primitive-action space over each agent's primitive-action set $A_i$; $M = \times_{i \in I} M_i$ is the joint macro-action space over each agent's macro-action space $M_i$; $\Omega = \times_{i \in I} \Omega_i$ is the joint primitive-observation space over each agent's primitive-observation set $\Omega_i$; $\zeta = \times_{i \in I} \zeta_i$ is the joint macro-observation space over each agent's macro-observation space $\zeta_i$; $T(s, \vec{a}, s') = P(s'|s, \vec{a})$ is the environmental transition dynamics; and $R(s, \vec{a})$ is a global reward function. During execution, each agent independently selects a macro-action $m_i$ using a high-level policy $\Psi_i : H_i^M \times M_i \to [0, 1]$ and captures a macro-observation $z_i \in \zeta_i$ according to the macro-observation probability function $Z_i(z_i, m_i, s') = P(z_i \mid m_i, s')$ when the macro-action terminates in a state $s'$. Each macro-action is represented as $m_i = \langle I_{m_i}, \pi_{m_i}, \beta_{m_i} \rangle$, where the initiation set $I_{m_i} \subset H_i^M$ defines how to initiate a macro-action based on macro-observation-action history $H_i^M$ at the high-level; $\pi_{m_i} : H_i^A \times A_i \to [0, 1]$ is the low-level policy for achieving a macro-action, and during the running, the agent receives a primitive-observation $o_i \in \Omega_i$ based on the observation function $O_i(o_i, a_i, s) = P(o_i|a_i, s)$ at every time step; $\beta_{m_i} : H_i^A \to [0, 1]$ is a stochastic termination function that determines how to terminate a macro-action based on primitive-observation-action history $H_i^A$ at the low-level. The objective of solving MacDec-POMDPs with finite horizon is to find a joint high-level policy $\vec{\Psi} = \times_{i \in I} \Psi_i$ that maximizes the value, $V^{\vec{\Psi}}(s_{(0)}) = \mathbb{E}\left[ \sum_{t=0}^{\mathbb{H}-1} \gamma^t r(s_{(t)}, \vec{a}_{(t)}) \mid s_{(0)}, \vec{\pi}, \vec{\Psi} \right]$, where $\gamma \in [0, 1]$ is a discount factor, and $\mathbb{H}$ is the number of (primitive) time steps until the problem terminates (the horizon).

## 2.2 Single-Agent Actor-Critic

In single-agent reinforcement learning, the *policy gradient theorem* [Sutton et al., 2000] formulates a principled way to optimize a parameterized policy $\pi_\theta$ via gradient ascent on the policy's performance defined as $J(\theta) = \mathbb{E}_{\pi_\theta}\left[ \sum_{t=0}^{\infty} \gamma^t r(s_{(t)}, a_{(t)}) \right]$. In POMDPs, the gradient w.r.t. parameters of a history-based policy $\pi_\theta(a \mid h)$ is expressed as: $\nabla_\theta J(\theta) = \mathbb{E}_{\pi_\theta}\left[ \nabla_\theta \log \pi_\theta(a \mid h) Q^{\pi_\theta}(h, a) \right]$, where $h$ is maintained by a recurrent neural network (RNN) [Hausknecht and Stone, 2015]. The actor-critic framework [Konda and Tsitsiklis, 2000] learns an on-policy action-value function $Q_\phi^{\pi_\theta}(h, a)$ (critic) via *temporal-difference* (TD) learning [Sutton, 1988] to approximate the action-value for the policy (actor) updates. Variance reduction is commonly achieved by training a history-value function $V_{\mathbf{w}}^{\pi_\theta}(h)$ and using it as a baseline [Weaver and Tao, 2001] as well as bootstrapping to estimate the action-value. Accordingly, the policy gradient can be written as:

$$\nabla_\theta J(\theta) = \mathbb{E}_{\pi_\theta}\left[ \nabla_\theta \log \pi_\theta(a \mid h)\big(r + \gamma V_{\mathbf{w}}^{\pi_\theta}(h') - V_{\mathbf{w}}^{\pi_\theta}(h)\big) \right] \tag{1}$$

where, $r$ is the immediate reward received by the agent at the corresponding time step.

## 2.3 Independent Actor-Critic

The single-agent actor-critic algorithm can be adapted to multi-agent problems in a simple way such that each agent independently learns its own actor and critic while treating other agents as part of the world [Foerster et al., 2018]. We consider a variance reduction version of *independent actor-critic* (IAC) with the policy gradient as follows:

$$\nabla_{\theta_i} J(\theta_i) = \mathbb{E}_{\vec{\pi}_{\vec{\theta}}}\left[ \nabla_{\theta_i} \log \pi_{\theta_i}(a_i|h_i)\big(r + \gamma V_{\mathbf{w}_i}^{\pi_{\theta_i}}(h_i') - V_{\mathbf{w}_i}^{\pi_{\theta_i}}(h_i)\big) \right] \tag{2}$$

where, $r$ is a shared reward over agents at every time step. Due to other agents' policy updating and exploring, from any agent's local perspective, the environment appears non-stationary which can lead to unstable learning of the critic without convergence guarantees [Lowe et al., 2017]. This instability often prevents IAC from learning high-quality cooperative policies.

## 2.4 Independent Actor with Centralized Critic

To address the above difficulties in independent learning approaches, centralized training for decentralized execution (CTDE) provides agents with access to global information during offline training while allowing agents to rely on only local information during online decentralized execution. Typically, the key idea of exploiting CTDE with actor-critic is to train a joint action-value function, $Q_\phi^{\vec{\pi}_{\vec{\theta}}}(\mathbf{x}, \vec{a})$, as the centralized critic and use it to compute gradients w.r.t. the parameters of each decentralized

policy [Foerster et al., 2018, Lowe et al., 2017]. Although the centralized critic can facilitate the update of decentralized policies to optimize global collaborative performance, it also introduces extra variance over other agents' actions [Lyu et al., 2021, Wang et al., 2021a]. Therefore, we consider the version of *independent actor with centralized critic* (IACC) with a general variance reduction trick [Foerster et al., 2018, Su et al., 2021], the policy gradient of which is:

$$\nabla_{\theta_i} J(\theta_i) = \mathbb{E}_{\vec{\pi}_{\vec{\theta}}}\Big[\nabla_{\theta_i} \log \pi_{\theta_i}(a_i \mid h_i)\big(r + \gamma V_{\mathbf{w}}^{\vec{\pi}_{\vec{\theta}}}(\mathbf{x}') - V_{\mathbf{w}}^{\vec{\pi}_{\vec{\theta}}}(\mathbf{x})\big)\Big] \tag{3}$$

where, $\mathbf{x}$ represents the available centralized information (e.g., joint observation, joint observation-action history, or the true state).

## 2.5 Learning Macro-Action-Based Deep Q-Nets

Previous MARL methods for Dec-POMDPs cannot work with the asynchronicity of macro-action-based agents, where agents may start and complete their macro-actions at different time steps. Recently, macro-action-based multi-agent DQNs have been proposed for MacDec-POMDPs [Xiao et al., 2019].

For decentralized learning, a new buffer, *Macro-Action Concurrent Experience Replay Trajectories* (Mac-CERTs), is designed for collecting each agent's macro-observation, macro-action, and reward information. In this buffer, the transition experience of each agent $i$ is represented as a tuple $\langle z_i, m_i, z_i', r_i^c \rangle$, where $r_i^c = \sum_{t=t_{m_i}}^{t_{m_i}+\tau_{m_i}-1} \gamma^{t-t_{m_i}} r_{(t)}$ is a cumulative reward of the macro-action taking $\tau_{m_i}$ time steps to be completed from its beginning time step $t_{m_i}$. During training, a mini-batch of concurrent sequential experiences is sampled from Mac-CERTs. Each agent independently accesses its own sampled experiences and obtains 'squeezed' trajectories by removing the transitions in the middle of each macro-action execution, which produces a mini-batch of transitions when the corresponding macro-action terminates (i.e., removing time information). Updates for each macro-action-value function $Q_{\phi_i}(h_i, m_i)$ take place only when the agent's macro-action is complete by minimizing a TD loss over the 'squeezed' data. In the centralized learning case, the objective is to learn a joint macro-action-value function $Q_\phi(\vec{h}, \vec{m})$. To this end, another special buffer called *Macro-Action Joint Experience Replay Trajectories* (Mac-JERTs) is developed for collecting agents' joint transition experience at every time step and each is represented as a tuple $\langle \vec{z}, \vec{m}, \vec{z}', \vec{r}^c \rangle$, where $\vec{r}^c = \sum_{t=t_{\vec{m}}}^{t_{\vec{m}}+\tau_{\vec{m}}-1} \gamma^{t-t_{\vec{m}}} r_{(t)}$ is a shared joint cumulative reward from the beginning time step $t_{\vec{m}}$ of the joint macro-action $\vec{m}$ to its termination, defined as when *any* agent finishes its own macro-action, after $\vec{\tau}_{\vec{m}}$ time steps. In each training iteration, the joint macro-action-value function is optimized over a mini-batch of 'squeezed' (depending on each joint macro-action termination) sequential joint experiences via TD learning. Other choices for what information to retain are also possible (e.g., the whole sequence of macro-actions or including time to complete) but this squeezing procedure was found to work well. In our macro-action-based actor-critic methods, we extend the above approaches to train critics on-policy, and the trajectory squeezing is changed variously for each method in order to achieve improved asynchronous macro-action-based policy updates via policy gradient.

## 3 Approach

MARL with asynchronous macro-actions is more challenging as it is difficult to determine *when* to update each agent's policy and *what* information to use. Although the macro-action-based DQN methods [Xiao et al., 2019] (in Section 2.5) give us the base to learn macro-action value functions, they do not directly extend to the policy gradient case, particularly in the case of centralized training for decentralized execution (CTDE). In this section, we propose principled formulations of on-policy macro-action-based multi-agent actor-critic methods for decentralized learning (Section 3.1), centralized learning (Section 3.2), and CTDE (Section 3.3). In each case, we first introduce the version with a Q-value function as the critic and then present the variance reduction version [2].

## 3.1 Macro-Action-Based Independent Actor-Critic (Mac-IAC)

Similar to the idea of IAC with primitive-actions (Section 2.3), a straightforward extension is to have each agent independently optimize its own macro-action-based policy (actor) using a local

---

[2]We use $h_i$ to represent an agent's local macro-observation-action history, and $\vec{h}$ to represent the joint history.

macro-action-value function (critic). Hence, we start with deriving a *macro-action-based policy gradient theorem* in Appendix B by incorporating the general Bellman equation for the state values of a macro-action-based policy [Sutton et al., 1999] into the *policy gradient theorem* in MDPs [Sutton et al., 2000], and then extend it to MacDec-POMDPs so that each agent can have the following policy gradient w.r.t. the parameters of its macro-action-based policy $\Psi_{\theta_i}(m_i|h_i)$ as: $\nabla_{\theta_i} J(\theta_i) = \mathbb{E}_{\vec{\Psi}_{\vec{\theta}}}\left[\nabla_{\theta_i} \log \Psi_{\theta_i}(m_i \mid h_i) Q_{\phi_i}^{\Psi_{\theta_i}}(h_i, m_i)\right]$. During training, each agent accesses its own trajectories and squeezes them in the same way as the decentralized case mentioned in Section 2.5 to train the critic $Q_{\phi_i}^{\Psi_{\theta_i}}(h_i, m_i)$ via on-policy TD learning and perform gradient ascent to update the policy when the agent's macro-action terminates. In our case, we train a local history value function $V_{\mathbf{w}_i}^{\Psi_{\theta_i}}(h_i)$ as each agent's critic and use it as a baseline to achieve variance reduction. The corresponding policy gradient is as follows:

$$\nabla_{\theta_i} J(\theta_i) = \mathbb{E}_{\vec{\Psi}_{\vec{\theta}}}\left[\nabla_{\theta_i} \log \Psi_{\theta_i}(m_i \mid h_i)\left(r_i^c + \gamma^{\tau_{m_i}} V_{\mathbf{w}_i}^{\Psi_{\theta_i}}(h_i') - V_{\mathbf{w}_i}^{\Psi_{\theta_i}}(h_i)\right)\right] \tag{4}$$

where, the cumulative reward $r_i^c$ is w.r.t. the execution of agent $i$'s macro-action $m_i$.

## 3.2 Macro-Action-Based Centralized Actor-Critic (Mac-CAC)

In the fully centralized learning case, we treat all agents as a single joint agent to learn a centralized actor $\Psi_\theta(\vec{m} \mid \vec{h})$ with a centralized critic $Q_\phi^{\Psi_\theta}(\vec{h}, \vec{m})$, and the policy gradient can be expressed as: $\nabla_\theta J(\theta) = \mathbb{E}_{\Psi_\theta}\left[\nabla_\theta \log \Psi_\theta(\vec{m} \mid \vec{h}) Q_\phi^{\Psi_\theta}(\vec{h}, \vec{m})\right]$. Similarly, to achieve a lower variance optimization for the actor, we learn a centralized history value function $V_{\mathbf{w}}^{\Psi_\theta}(\vec{h})$ by minimizing a TD-error loss over joint trajectories that are squeezed w.r.t. each joint macro-action termination (when *any* agent terminates its macro-action, defined in the centralized case in Section 2.5). Accordingly, the policy's updates are performed when each joint macro-action is completed by ascending the following gradient:

$$\nabla_\theta J(\theta) = \mathbb{E}_{\Psi_\theta}\left[\nabla_\theta \log \Psi_\theta(\vec{m} \mid \vec{h})\left(\vec{r}^c + \gamma^{\vec{\tau}_{\vec{m}}} V_{\mathbf{w}}^{\Psi_\theta}(\vec{h}') - V_{\mathbf{w}}^{\Psi_\theta}(\vec{h})\right)\right] \tag{5}$$

where the cumulative reward $\vec{r}^c$ is w.r.t. the execution of the joint macro-action $\vec{m}$.

## 3.3 Macro-Action-Based Independent Actor with Centralized Critic (Mac-IACC)

As mentioned earlier, fully centralized learning requires perfect online communication that is often hard to guarantee, and fully decentralized learning suffers from environmental non-stationarity due to agents' changing policies. In order to learn better decentralized macro-action-based policies, in this section, we propose two macro-action-based actor-critic algorithms using the CTDE paradigm. The difference between a joint macro-action termination from the centralized perspective and a macro-action termination from each agent's local perspective gives rise to a new challenge: *what kind of centralized critic should be learned and how should it be used to optimize decentralized policies where some have completed and some have not*, which we investigate below.

**Naive Mac-IACC.** A naive way of incorporating macro-actions into a CTDE-based actor-critic framework is to directly adapt the idea of the primitive-action-based IACC (Section 2.4) to have a shared joint macro-action-value function $Q_\phi^{\vec{\Psi}_{\vec{\theta}}}(\mathbf{x}, \vec{m})$ in each agent's decentralized macro-action-based policy gradient as: $\nabla_{\theta_i} J(\theta_i) = \mathbb{E}_{\vec{\Psi}_{\vec{\theta}}}\left[\nabla_{\theta_i} \log \Psi_{\theta_i}(m_i \mid h_i) Q_\phi^{\vec{\Psi}_{\vec{\theta}}}(\mathbf{x}, \vec{m})\right]$. To reduce variance, with a value function $V_{\mathbf{w}}^{\vec{\Psi}_{\vec{\theta}}}(\mathbf{x})$ as the centralized critic, the policy gradient w.r.t. the parameters of each agent's high-level policy can be rewritten as:

$$\nabla_{\theta_i} J(\theta_i) = \mathbb{E}_{\vec{\Psi}_{\vec{\theta}}}\left[\nabla_{\theta_i} \log \Psi_{\theta_i}(m_i \mid h_i)\left(\vec{r}^c + \gamma^{\vec{\tau}_{\vec{m}}} V_{\mathbf{w}}^{\vec{\Psi}_{\vec{\theta}}}(\mathbf{x}') - V_{\mathbf{w}}^{\vec{\Psi}_{\vec{\theta}}}(\mathbf{x})\right)\right] \tag{6}$$

Here, the critic is trained in the fully centralized manner described in Section 3.2 while allowing it to access additional global information (e.g., joint macro-observation-action history, ground truth

state or both) represented by the symbol $\mathbf{x}$. However, updates of each agent's policy $\Psi_{\theta_i}(m_i \mid h_i)$ only occur at the agent's own macro-action termination time steps rather than depending on joint macro-action terminations in the centralized critic training.

**Independent Actor with Individual Centralized Critic (Mac-IAICC).** Note that naive Mac-IACC is technically incorrect. The cumulative reward $\vec{r}^{\,c}$ in Eq. 6 is based on the corresponding joint macro-action's termination that is defined as when *any* agent finishes its own macro-action, which produces two potential issues: a) $\vec{r}^{\,c} + \gamma^{\vec{\tau}_{\vec{m}}} V_{\mathbf{w}}^{\vec{\Psi}_{\vec{\theta}}}(\mathbf{x}')$ may not estimate the value of the macro-action $m_i$ well as the reward does not depend on $m_i$'s termination; b) from agent $i$'s perspective, its policy gradient estimation may involve higher variance associated with the asynchronous macro-action terminations of other agents.

To tackle the aforementioned issues, we propose to learn a separate centralized critic $V_{\mathbf{w}_i}^{\vec{\Psi}_{\vec{\theta}}}(\mathbf{x}')$ for each agent via TD-learning. In this case, the TD-error for updating $V_{\mathbf{w}_i}^{\vec{\Psi}_{\vec{\theta}}}(\mathbf{x}')$ is computed by using the reward $r_i^c$ that is accumulated purely based on the execution of the agent $i$'s macro-action $m_i$. With this TD-error estimation, each agent's decentralized macro-action-based policy gradient becomes:

$$\nabla_{\theta_i} J(\theta_i) = \mathbb{E}_{\vec{\Psi}_{\vec{\theta}}}\left[ \nabla_{\theta_i} \log \Psi_{\theta_i}(m_i \mid h_i)\left( r_i^c + \gamma^{\tau_{m_i}} V_{\mathbf{w}_i}^{\vec{\Psi}_{\vec{\theta}}}(\mathbf{x}') - V_{\mathbf{w}_i}^{\vec{\Psi}_{\vec{\theta}}}(\mathbf{x}) \right) \right] \tag{7}$$

Now, from agent $i$'s perspective, $r_i^c + \gamma^{\tau_{m_i}} V_{\mathbf{w}_i}^{\vec{\Psi}_{\vec{\theta}}}(\mathbf{x}')$ is able to offer a more accurate value prediction for the macro-action $m_i$, since both the reward, $r_i^c$ and the value function $V_{\mathbf{w}_i}^{\vec{\Psi}_{\vec{\theta}}}(\mathbf{x}')$ depend on agent $i$'s macro-action termination. Also, unlike the case in Naive Mac-IACC, other agents' terminations cannot lead to extra noisy estimated rewards w.r.t. $m_i$ anymore so that the variance on policy gradient estimation gets reduced. Then, updates for both the critic and the actor occur when the corresponding agent's macro-action ends and take the advantage of information sharing. The pseudocode and detailed trajectory squeezing process for each proposed method are presented in Appendix C.

## 4 Simulation Experiments

### 4.1 Domain Setup

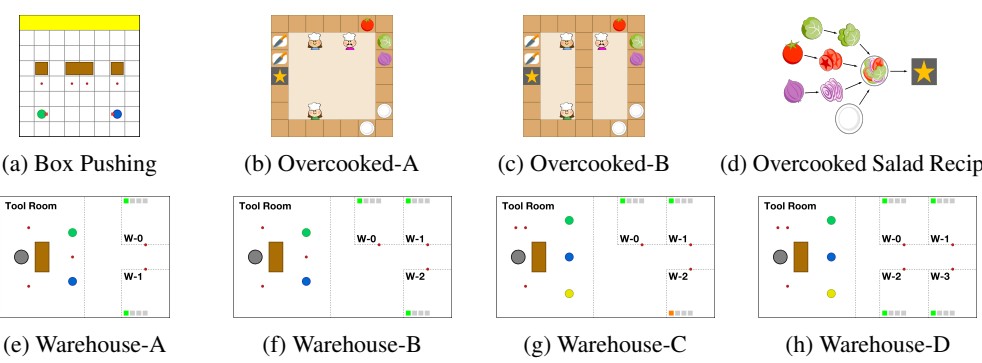

(a) Box Pushing    (b) Overcooked-A    (c) Overcooked-B    (d) Overcooked Salad Recipe

(e) Warehouse-A    (f) Warehouse-B    (g) Warehouse-C    (h) Warehouse-D

Figure 1: Experimental environments.

We investigate the performance of our algorithms over a variety of multi-agent problems with macro-actions (Fig. 1): Box Pushing [Xiao et al., 2019], Overcooked [Wu et al., 2021b], and a larger Warehouse Tool Delivery [Xiao et al., 2019] domain. Macro-actions are defined by using prior domain knowledge as they are straightforward in these tasks. Typically, we also include primitive-actions into macro-action set (as one-step macro-actions), which gives agents the chance to learn more complex policies that use both when it is necessary. We describe the domains' key properties here and have more details in Appendix D.

**Box Pushing** (Fig. 1a). The optimal solution for the two agents is to cooperatively push the big box to the yellow goal area for a terminal reward, but partial observability makes this difficult. Specifically, robots have four primitive-actions: *move forward*, *turn-left*, *turn-right* and *stay*. In the macro-action

case, each robot has three one-step macro-actions: ***Turn-left***, ***Turn-right***, and ***Stay***, as well as three multi-step macro-actions: ***Move-to-small-box(i)*** and ***Move-to-big-box(i)*** navigate the robot to the red spot below the corresponding box and terminate with the robot facing the box; ***Push*** causes the robot to keep moving forward until arriving at the world's boundary (potentially pushing the small box or trying to push the big one). The big box only moves if both agents push it together. Each robot can only observe the status (*empty*, *teammate*, *boundary*, *small or big box*) of the cell in front of it. A penalty is issued when any robot hits the boundary or pushes the big box alone.

**Overcooked** (Fig. 1b - 1c). Three agents must learn to cooperatively prepare a lettuce-tomato-onion salad and deliver it to the 'star' cell. The challenge is that the salad's recipe (Fig. 1d) is unknown to agents. With primitive-actions (*move up*, *down*, *left*, *right*, and *stay*), agents can move around and achieve picking, placing, chopping and delivering by standing next to the corresponding cell and moving against it (e.g., in Fig. 1b, the pink agent can *move right* and then *move up* to pick up the tomato). We describe the major function of macro-actions below and full details (e.g., termination conditions) are included in Appendix D.2. Each agent's macro-action set consists of: a) five one-step macro-actions that are the same as the primitive ones; b) ***Chop***, cuts a raw vegetable into pieces when the agent stands next to a cutting board and an unchopped vegetable is on the board, otherwise it does nothing; c) long-term navigation macro-actions: ***Get-Lettuce***, ***Get-Tomato***, ***Get-Onion***, ***Get-Plate-1/2***, ***Go-Cut-Board-1/2*** and ***Deliver***, which navigate the agent to the location of the corresponding object with various possible terminal effects (e.g., holding a vegetable in hand, placing a chopped vegetable on a plate, arriving at the cell next to a cutting board, delivering an item to the star cell, or immediately terminating when any property condition does not hold, e.g., no path is found or the vegetable/plate is not found); d) ***Go-Counter*** (only available in Overcook-B, Fig. 1c), navigates an agent to the center cell in the middle of the map when the cell is not occupied, otherwise, it moves to an adjacent cell. If the agent is holding an object or one is at the cell, the object will be placed or picked up. Each agent only observes the *positions* and *status* of the entities within a $5 \times 5$ square centered on the robot.

**Warehouse Tool Delivery** (Fig. 1e - 1h). In each workshop (e.g., W-0), a human is working on an assembly task (involving 4 sub-tasks that each takes a number of time steps to complete) and requires three different tools for future sub-tasks to continue. A robot arm (grey) must find tools for each human on the table (brown) and pass them to mobile robots (green, blue and yellow) who are responsible for delivering tools to humans. Note that, the correct tools needed by each human are unknown to robots, which has to be learned during training in order to perform efficient delivery. A delayed delivery leads to a penalty. We consider variants with two or three mobile robots and two to four humans to examine the scalability of our methods (Fig. 1f - 1h). We also consider one faster human (orange) to check if robots can prioritize him (Fig. 1g). Mobile robots have the following macro-actions: ***Go-W(i)***, moves to the waypoint (red) at workshop $i$; ***Go-TR***, goes to the waypoint at the right side of the tool room (covered by the blue robot in Fig. 1g and 1h); and ***Get-Tool***, navigates to a pre-allocated waypoint (that is different for each robot to avoid collisions) next to the robot arm and waits there until either receiving a tool or 10 time steps have passed. The robot arm's applicable macro-actions are: ***Search-Tool(i)***, finds tool $i$ and places it in a staging area (containing at most two tools) on the table, and otherwise, it freezes the robot for the amount of time the action would take when the area is fully occupied; ***Pass-to-M(i)***, passes the first staged tool to mobile robot $i$; and ***Wait-M***, waits for 1 time step. The robot arm only observes the *type* of each tool in the staging area and *which mobile robot* is waiting at the adjacent waypoints. Each mobile robot always knows its *position* and the *type* of tool that it is carrying, and can observe the *number* of tools in the staging area or the *sub-task* a human is working on only when at the tool room or the workshop respectively.

## 4.2 Results and Discussions

We evaluate performance of one training trial with a mean discounted return measured by periodically (every 100 episodes) evaluating the learned policies over 10 testing episodes. We plot the average performance of each method over 20 independent trials with one standard error and smooth the curves over 10 neighbors. We also show the optimal expected return in Box Pushing domain as a dash-dot line. More training details are in Appendix E.

**Advantages of learning with macro-actions**. We first present a comparison of our macro-action-based actor-critic methods against the primitive-action-based methods in fully decentralized and fully centralized cases. We consider various grid world sizes of the Box Pushing domain (top row in Fig. 2) and two Overcooked scenarios (bottom row in Fig. 2). The results show significant performance

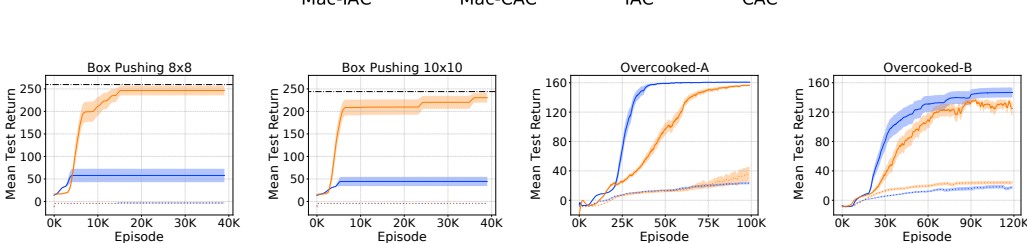

Figure 2: Decentralized learning and centralized learning with macro-actions vs primitive-actions.

improvements of using macro-actions over primitive-actions. More concretely, in the Box Pushing domain, reasoning about primitive movements at every time step makes the problem intractable so the robots cannot learn any good behaviors in primitive-action-based approaches other than to keep moving around. Conversely, Mac-CAC reaches near-optimal performance, enabling the robots to push the big box together. Unlike the centralized critic which can access joint information, even in the macro-action case, it is hard for each robot's decentralized critic to correctly measure the responsibility for a penalty caused by a teammate pushing the big box alone. Mac-IAC thus converges to a local-optima of pushing two small boxes in order to avoid getting the penalty.

In the Overcooked domain, an efficient solution requires the robots to asynchronously work on independent subtasks (e.g., in scenario A, one robot gets a plate while another two robots pick up and chop vegetables; and in scenario B, the right robot transports items while the left two robots prepare the salad). This large amount of independence explains why Mac-IAC can solve the task well. This also indicates that using local information is enough for robots to achieve high-quality behaviors. As a result, Mac-CAC learns slower because it must figure out the redundant part of joint information in much larger joint macro-level history and action spaces than the spaces in the decentralized case. The primitive-action-based methods begin to learn, but perform poorly in such long-horizon tasks.

**Advantages of having individual centralized critics.** Fig. 3 shows the evaluation of our methods in all three domains. As each agent's observation is extremely limited in Box Pushing, we allow centralized critics in both Mac-IAICC and Naive Mac-IACC to access the state (agents' poses and boxes' positions), but use the joint macro-observation-action history in the other two domains.

In the Box Pushing task (the left two in the top row in Fig. 3), Naive Mac-IACC (green) can learn policies almost as good as the ones for Mac-IAICC (red) for the smaller domain, but as the grid world size grows, Naive Mac-IACC performs poorly while Mac-IAICC keeps its performance near the centralized approach. From each agent's perspective, the bigger the world size is, the more time steps a macro-action could take, and the less accurate the critic of Naive Mac-IACC becomes since it is trained depending on any agent's macro-action termination. Conversely, Mac-IAICC gives each agent a separate centralized critic trained with the reward associated with its own macro-action execution.

In Overcooked-A (the third one at the top row in Fig. 3), as Mac-IAICC's performance is determined by the training of three agents' critics, it learns slower than Naive Mac-IACC in the early stage but converges to a slightly higher value and has better learning stability than Naive Mac-IACC in the end. The result of scenario B (the last one at the top row in Fig. 3) shows that Mac-IAICC outperforms other methods in terms of achieving better sample efficiency, a higher final return and a lower variance. The middle wall in scenario B limits each agent's moving space and leads to a higher frequency of macro-action terminations. The shared centralized critic in Naive Mac-IACC thus provides more noisy value estimations for each agent's actions. Because of this, Naive Mac-IACC performs worse with more variance. Mac-IAICC, however, does not get hurt by such environmental dynamics change. Both Mac-CAC and Mac-IAC are not competitive with Mac-IAICC in this domain.

In the Warehouse scenarios (the bottom row in Fig. 3), Mac-IAC (blue) performs the worst due to its natural limitations and the domain's partial observability. In particular, it is difficult for the gray robot (arm) to learn an efficient way to find the correct tools purely based on local information and very delayed rewards that depend on the mobile robots' behaviors. In contrast, in the fully centralized Mac-CAC (orange), both the actor and the critic have global information so it can learn faster in the early training stage. However, Mac-CAC eventually gets stuck at a local-optimum in all five scenarios

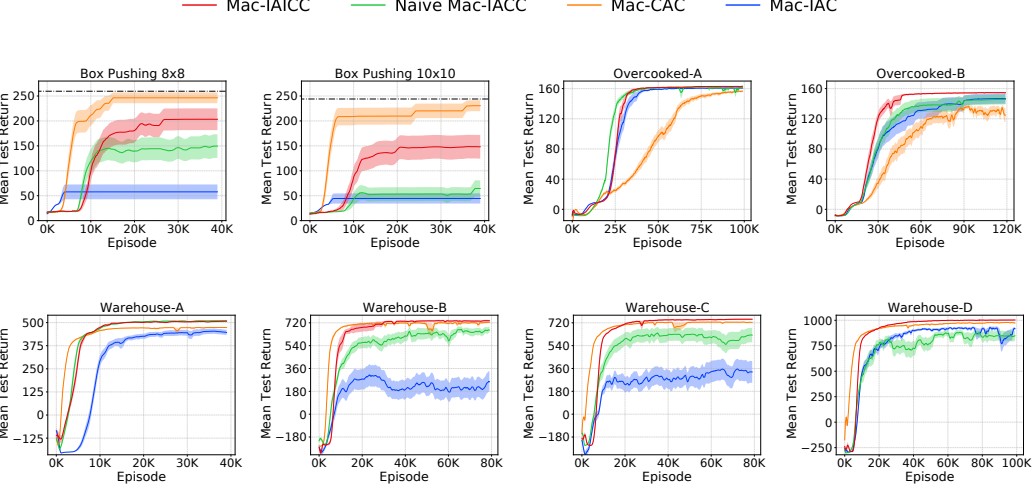

Figure 3: Comparison of macro-action-based asynchronous actor-critic methods.

due to the exponential dimensionality of joint history and action spaces over robots. By leveraging the CTDE paradigm, both Mac-IAICC and Naive Mac-IACC perform the best in warehouse A. Yet, the weakness of Naive Mac-IACC is clearly exposed when the problem is scaled up in Warehouse B, C and D. In these larger cases, the robots' asynchronous macro-action executions (e.g., traveling between rooms) become more complex and cause more mismatching between the termination from each agent's local perspective and the termination from the centralized perspective, and therefore, Naive Mac-IACC's performance significantly deteriorates, even getting worse than Mac-IAC in Warehouse-D. In contrast, Mac-IAICC can maintain its outstanding performance, converging to a higher value with much lower variance, compared to other methods. This outcome confirms not only Mac-IAICC's scalability but also the effectiveness of having an individual critic for each agent to handle variable degrees of asynchronicity in agents' high-level decision-making.

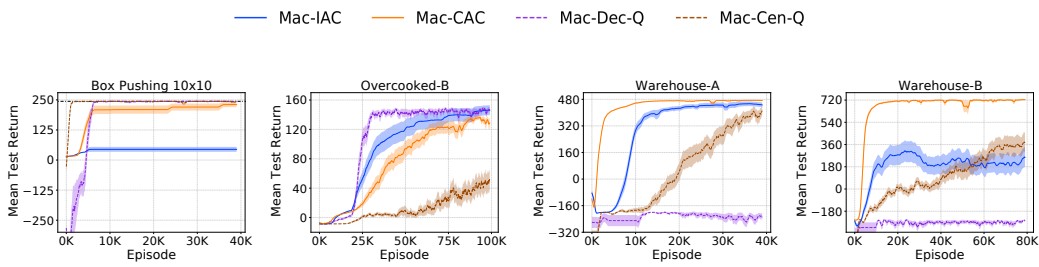

Figure 4: Comparisons of macro-action-based actor-critic methods and value-based methods.

**Comparative analysis between actor-critic and value-based approaches**. We also compare our actor-critic methods (Mac-IAC and Mac-CAC) with the current state-of-the-art asynchronous decentralized and centralized MARL methods, the value-based approaches (Mac-Dec-Q and Mac-Cen-Q) [Xiao et al., 2019], shown in Fig. 4. The Box Pushing task requires agents to simultaneously reach the big box and push it together. This consensus is rarely achieved when agents independently sample actions using stochastic policies in Mac-IAC and is hard to learn from pure on-policy data. By having a replay-buffer, value-based approaches show much stronger sample efficiency than on-policy actor-critic approaches in this domain with a small action space (left figure). Such an advantage is sustained by the decentralized value-based method (Mac-Dec-Q) but gets lost in the centralized one (Mac-Cen-Q) in the Overcooked domains due to a huge joint macro-action space ($15^3$). On the contrary, our actor-critic methods can scale to large domains and learn high-quality solutions. This is particularly noticeable on Warehouse-A, where the policy gradient methods quickly learn a high-

quality policy while the centralized Mac-Cen-Q is slow to learn and the decentralized Mac-Dec-Q is unable to learn. In addition, the stochastic policies in actor-critic methods potentially have better exploration property so that, in Warehouse domains, Mac-IAC can bypass an obvious local-optima that Mac-Dec-Q falls into, where the robot arm greedily chooses *Wait-M* to avoid more penalties.

## 5    Hardware Experiments

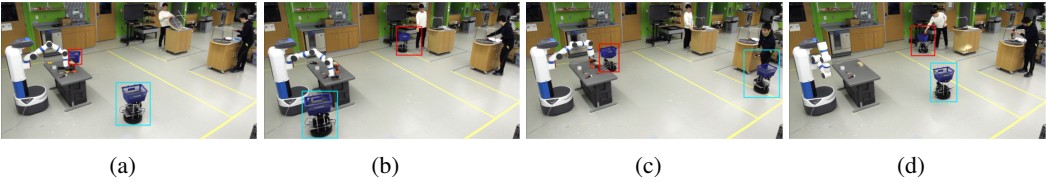

| (a) | (b) | (c) | (d) |

Figure 5: Collaborative behaviors generated by running the decentralized policies learned by Mac-IAICC where Turtlebot-0 (T-0) is bounded in red and Turtlebot-1 (T-1) is bounded in blue. (a) After staging a tape measure at the left, Fetch looks for the 2nd one while Turtlebots approach the table; (b) T-0 deliveries a tap measure to W-0 and T-1 waits for a clamp from Fetch; (c) T-1 deliveries a clamp to W-1, while T-0 carries the other clamp and goes to W-0, and Fetch searches for an electric drill; (d) T-0 deliveries an electric drill (the last tool) to W-0 and the entire delivery task is completed.

We also extend scenario A of the Warehouse Tool Delivery task to a hardware domain (details of experimental setup are referred to Appendix F). Fig. 5 shows the sequential collaborative behaviors of the robots in one hardware trial. Fetch was able to find tools in parallel such that two tape measures (Fig. 5a), two clamps (Fig. 5b) and two electric drills, were found instead of finding all three types of tool for one human and then moving on to the other which would result in one of the humans waiting. Fetch's efficiency is also reflected in the behaviors such that it passed a tool to the Turtelbot who arrived first (Fig. 5b) and continued to find the next tool when there was no Turtlebot waiting beside it (Fig. 5c). Meanwhile, Turtlebots were clever such that they successfully avoid delayed delivery by sending tools one by one to the nearby workshop (e.g., T-0 focused on W-0 shown in Fig. 5b and 5d, and T-1 focused on W-1 shown in Fig. 5c), rather than waiting for all tools before delivering, traveling a longer distance to serve the human at the diagonal, or prioritizing one of the humans altogether.

## 6    Conclusion

This paper introduces a general formulation for asynchronous multi-agent macro-action-based policy gradients under partial observability along with proposing a decentralized actor-critic method (Mac-IAC), a centralized actor-critic method (Mac-CAC), and two CTDE-based actor-critic methods (Naive Mac-IACC and Mac-IAICC). These are the first approaches to be able to incorporate controllers that may require different amounts of time to complete (macro-actions) in a general asynchronous multi-agent actor-critic framework. Empirically, our methods are able to learn high-quality macro-action-based policies allowing agents to perform asynchronous collaborations in large and long-horizon problems. Importantly, our most advanced method, Mac-IAICC, allows agents to have individual centralized critics tailored to the agent's own macro-action execution. Additionally, the practicality of our approach is validated in a real-world multi-robot setup based on a warehouse domain. This work provides a foundation for future macro-action-based MARL algorithm development, including other policy gradient-based methods as well as methods which also learn the macro-actions.

## Acknowledgments

We thank Chengguang Xu and Tian Xia for their participation in hardware experiments. This research is supported in part by the U.S. Office of Naval Research under award number N00014-19-1-2131, Army Research Office award W911NF20-1-0265 and NSF CAREER Award 2044993.

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
