# OpenReview forum: "Asynchronous Actor-Critic for Multi-Agent Reinforcement Learning"
_NeurIPS.cc/2022/Conference — NeurIPS 2022 Accept_

### Official Review · Reviewer_vcLk · 2022-07-08

**Rating:** 5
**Confidence:** 3
**Soundness:** 3 good
**Presentation:** 3 good
**Contribution:** 2 fair

**Summary:**

This paper extends macro-action-based DQN to actor-critic methods to solve asynchronous hierarchical multi-agent cooperation tasks. Asynchronous learning, where the high-level decisions have different action durations, is an interesting and important topic for complex real-world applications. The motivation is clear, and the paper is well organized. The experiments are performed on both simulation environments and hardware platforms, which is convincing.

**Questions:**

The easiest method I can see to solve the duration problem is to set a maximal duration and set an idle action. Thus, the macro-actions have the same duration. What is the limitation of the easiest method?

Since the formulation is similar to the settings of hierarchical RL, I am curious about the performance of existing hierarchical MARL methods although they do not specifically consider the asynchronicity problem.

The hardware experiments are fancy. Although there is a video in Appendix, it is hard to intuitively get the effectiveness of the proposed methods from Figure 5.

**Limitations:**

The authors have discussed the limitations.

**Strengths And Weaknesses:**

Strengths:

Asynchronous learning is meaningful to many real-world applications.

The motivation is clear, and the paper is well organized.

The experiments are extensive and demonstrate that learning with macro-actions is important in tasks requiring high-level planning. And the selected platforms are suitable to verify the effectiveness of asynchronous learning.

I appreciate the section Mac-IAICC, where the duration issue is clearly discussed and specifically treated.

Weaknesses:

The novelty is limited since the proposed methods seem to be directly extended from macro-action-based DQN.

---

> ### Author Response · Authors · 2022-07-29
> **Response to Reviewer vcLk**
>
> We thank the reviewer for their time and comments.
>
> Regarding the novelty of this paper, we would like to highlight three points:
>
> - The extension from macro-action-based DQN to actor-critic is not trivial, especially for achieving centralized training for decentralized execution (CTDE) with macro-actions in an actor-critic framework, which is the major challenge addressed in this paper but is not considered at all in the value-based case. As we have mentioned in the paper (line186-189), it is hard to determine what the proper centralized critic would be for optimizing decentralized policies due to the inconsistency of macro-action execution between the decentralized perspective and the centralized perspective. This paper is the first work that detects and solves this significant issue by showing that a straightforward extension (Naive IACC) performs poorly across a variety of domains (Fig. 3), and proposing Mac-IAICC that can learn high-quality solutions.
>
> - This is the first work that formulates a set of actor-critic approaches for learning asynchronous macro-action-based policies in multi-agent settings.  The proposed decentralized, centralized and CTDE frameworks are all general formalisms, which allow the MARL community to build off of our contributions in this paper for future research on multi-agent asynchronous and hierarchical RL.
>
> - This is also the first work that deploys decentralized macro-action-based policies learned via policy gradients to efficiently solve a large real-world multi-robot task that the DQN-based approaches cannot solve well (refer to the results under warehouse scenarios in Fig. 4).
>
> We answer the reviewer’s specific questions below:
>
> Q1: Setting a maximal duration and an idle action is not possible in many settings and will significantly reduce the efficiency of the system in others. For example, many types of aerial and water vehicles cannot idle. Also, real-world robot systems often involve heterogeneous robots (e.g., aerial vehicles and ground vehicles) with very different speeds and (macro-)action types.  It is not clear what a reasonable maximal duration would be, especially in a learning system where there isn’t a model of the action interacting with the environment. Instead, in these systems, agents are naturally required to be able to perform asynchronous decision-making.
>
> Q2: Our setting assumes asynchronous learning and execution. As noted above, since the existing hierarchical MARL methods are synchronized they are not applicable to our general setting. When a max duration and an idle action can be used, it is of course possible to construct domains where such methods perform arbitrarily poorly.
>
> Q3: We agree with the reviewer and will add clarification in the final version with one extended page.

---

### Official Review · Reviewer_TCiE · 2022-07-09

**Rating:** 7
**Confidence:** 4
**Soundness:** 4 excellent
**Presentation:** 3 good
**Contribution:** 3 good

**Summary:**

This paper presents a general formulation of macro action-based Multi-Agent Actor Critic framework by extending widely used methods in multi-agent actor critic literature to a Mac-DecPOMDP setting with asynchronous actions. This is desirable since agents can now update themselves asynchronously without having to wait for other agents. Macro-actions are designed such that it can include both temporally extended actions, as well as, primitive actions.

Existing methods with macro-actions often rely on the assumption that agents behave synchronously and thus cannot be readily applied in the proposed setting. To this end, the authors propose a range of baselines (generalized from their primitive action counterparts) to compare with for evaluation purpose -- Mac-IAC, Mac-CAC, Naive IACC, and Mac-IAICC. The empirical results from three different environments and an additional real-life robotics application show significant performance and desirable behaviors in the agents.


**Questions:**

Q1. lines 74-75: Can the authors clarify how an agent's "primitive-observation set" is different from it's "macro-observation space". Isn't the latter a subset of the former? How does the environment have two different observation functions?

Q2. lines 302-303: Even after having access to the (full) state information in the Box Pushing task, not only does Mac-CAC outperform Mac-IAICC, it also learns faster. How is this possible given that both models have the same input and CAC has a huge joint macro-action space?

Minor Comments:

1. The authors should try to add more clarity as to why they had to design and compare with their own baselines (and not with existing methods on Mac-DecPOMDP). While I understand that existing methods cannot be readily generalized to the asynchronous setting (which the authors have mentioned in bits and pieces throughout the paper), issuing a more concrete statement upfront would be useful for the readers to place this work in the context of other similar works.

2. Lines 145-147: Please point to the appropriate sections in the Appendix.


**Limitations:**

In general, the authors have done well to highlight the limitations of the different proposed algorithms in Section 4.2. However, I felt that a general high-level discussion on the asynchronous updates in the algorithms is missing and the paper could benefit from placing this in the context of other potential approaches.

**Strengths And Weaknesses:**

Strengths:

1. Although the authors borrow the "trajectory squeezing" technique to achieve asynchronicity (as stated in Sec 2.5), the fact that it is adapted to an on-policy setting is pretty novel. Since, most practical applications in MARL display asynchronous behavior, the set of algorithms presented here are of significance to the MARL and the robotics community.

2. The paper has a clear flow and is well written. I, at least, did not come across any technical or grammatical errors (Kudos to the authors for that !)

3. The authors present their experimental results on three different environments each of which justify the use of one or another proposed macro-action paradigm (by showing clear improvements). Additionally, ablation studies were performed on a real-life robotic warehouse setting, which also clearly highlights desired behavioral aspects of the proposed methods. The authors have also submitted codes for reproduction (although I did not run them).

Weaknesses:

1. The "squeezing" process described in the main paper is a bit ambiguous, however, it is very clearly explained in the Appendix. Since the authors use this as an essential part of their training algorithm, perhaps it would be good to move the explanation from the Appendix to the main paper (just Figure 6 from Appendix would suffice).

2. Section 4.2 (Advantages of learning with macro-actions): This seems to be an unfair comparison given that macro-actions are pre-specified and it is obviously convenient for the macro-action-based agents to learn faster and better than primitive-action-based agents. As such, the results are not all that appealing. Perhaps this should be pushed to the Appendix to make room for explaining the algorithm (from point 1. above) in more detail.

3. The limitations of the proposed framework should be clearly highlighted. While the authors compare the shortcomings of each module in different environments, I felt that a general discussion of the algorithms performing asynchronous updates is missing. For instance, when you look at the trajectory squeezing process in the Appendix, is there any crucial information that the actor/critic misses out on during gradient updates, and if so, how can they be improved.

Please note that I am considering the above-mentioned weaknesses as minor issues and authors are requested to make changes to address these issues in the main paper.

---

> ### Author Response · Authors · 2022-07-29
> **Response to Reviewer TCiE**
>
> We thank the reviewer for their time and comments.
>
> In the final version, the paper will be allowed to have one page extension. We will certainly add clarification for the “squeezing” process and consider rearranging the results and discussion about the comparison between macro-action-based learners and primitive-action-based learners. Regarding squeezing, the process loses time information of the corresponding macro-action execution. While such information could potentially improve performance, we think it is not necessary in the proposed frameworks. The actor is a policy mapping from macro-observation space to macro-action space, therefore, the critic should provide an expected macro-action-value estimation over time in the policy gradient computation. Explicitly considering the time information may be helpful but would also lead to an increase in computation complexity. We will expand this limitation discussion in the final version.
>
> We answer the reviewer’s specific questions below:
>
> Q1: In the Box Pushing and Overcooked domains, each macro-observation includes the same features as the primitive one. As each agent receives a new macro-observation only when its macro-action terminates,  the macro-observation space ends up as a subset of the primitive one. However, in Warehouse problems, the two observation spaces are very different. For example, the low-level controller for manipulation has to include the angle, velocity, and acceleration of each joint of the robot arm as the primitive observation but this information is not needed in the macro-observation for learning high-level cooperative policies. As in the primitive case, choosing the observation set is important, but our proposed frameworks can work with any type of observation function. Using different observation (or action) sets may change the performance of the methods but we don’t expect such changes to affect the trends between the algorithms (i.e., the ranking of the algorithms in terms of performance should not change).
>
> Q2: In the Box Pushing domain, the number of joint macro-actions is $8^2=64$ which is not huge. Also, both frameworks actually have different inputs. In Mac-CAC, the centralized actor and critic can access joint information in both training and execution. In Mac-IAICC, although the critic can access the (full) state information during training, the execution is still decentralized based on only local information. Therefore, it is easy for Mac-CAC to explore good cooperative choices (e.g., simultaneously go to the big box and push it together) which has a high probability to be generated by the centralized actor during execution. In Mac-IAICC, the decentralized actors based on only local information would have a relatively lower probability of generating the above good cooperative behaviors, which slows the learning.
>
> Regarding the minor comments, we agree with the reviewer’s suggestions and will clarify them in the final version.

---

> > ### Comment · Reviewer_TCiE · 2022-08-04
> > **Reviewer TCiE Response to Rebuttal**
> >
> > Thank you for addressing the questions and comments. I am convinced with the author's arguments and would stick with my score.

---

### Official Review · Reviewer_6Bsf · 2022-07-11

**Rating:** 4
**Confidence:** 5
**Soundness:** 2 fair
**Presentation:** 3 good
**Contribution:** 2 fair

**Summary:**

This paper develops an asynchronous multi-agent actor-critic approach that allows agents to directly optimize asynchronous strategies in three state-of-the-art multi-agent training paradigms: decentralized learning, centralized learning, and decentralized execution and centralized training.This paper also evaluates its methods in both simulation and hardware environments. The experiment results show the method can obtain high-quality and asynchronous solutions.

**Questions:**

+ In the setting of this paper, the number, meaning and complexity of macro-actions are defined in advance. How much influence do different settings of macro-actions have on experiment results?

+ In line 343-346, the authors think the value-based approaches have better performance than actor-critic approaches because of their stronger sample efficiency. Why not compare with high sample-efficiency AC methods (such as MAPPO)?

**Limitations:**

The authors do not discuss limitations and potential negative societal impact of their work.

**Strengths And Weaknesses:**

Strengths:

+ The experimental design and discussion are very interesting. The experiments are implemented in both simulation and hardware environments, which demonstrates the powerful application ability of this method.
+ The statement is clear and easy-reading.


Weaknesses:

+ The idea of utilizing macro-action in MARL methods is not novel. This paper just transfers the macro-action-based value evaluation in DQN method[1] to AC method. To some extent, the paper lacks originality.


[1] Yuchen Xiao, Joshua Hoffman, and Christopher Amato. Macro-action-based deep multi-agent  reinforcement learning. In Proceedings of the Conference on Robot Learning, 2019.

---

> ### Author Response · Authors · 2022-07-29
> **Response to Reviewer 6Bsf**
>
> We thank the reviewer for their time and comments.
>
> Regarding the originality of this paper, we would like to highlight three points:
>
> - The extension from the value-based case [1] to actor-critic is not trivial, especially for achieving centralized training for decentralized execution (CTDE) with macro-actions in an actor-critic framework, which is the major challenge addressed in this paper but is not considered at all in [1]. As we have mentioned in the paper (line186-189), it is hard to determine what the proper centralized critic would be for optimizing decentralized policies due to the inconsistency of macro-action execution between the decentralized perspective and the centralized perspective. This paper is the first work that detects and solves this significant issue by showing that a straightforward extension (Naive IACC) performs poorly across a variety of domains (Fig. 3), and proposing Mac-IAICC that can learn high-quality solutions.
>
> - This is the first work that formulates a set of actor-critic approaches for learning asynchronous macro-action-based policies in multi-agent settings.  The proposed decentralized, centralized and CTDE frameworks are all general formalisms, which allow the MARL community to build off of our contributions in this paper for future research on multi-agent asynchronous and hierarchical RL.
>
> - This is also the first work that deploys decentralized macro-action-based policies learned via policy gradients to efficiently solve a large real-world multi-robot task that the DQN-based approaches in [1] cannot solve well (refer to the results under warehouse scenarios in Fig. 4).
>
> We answer the reviewer’s specific questions below:
>
> Q1: Investigating the influence of different settings of actions is an open question for both macro-action-based and primitive-action-based scenarios. For instance, in primitive-action-based multi-agent benchmark problems (e.g., OpenAI Particle Environments and SMAC), a set of primitive actions also has been predefined, and it is not clear about the potential impact of using a different set of actions. This topic is interesting but not the focus of this paper. Importantly, our methods are general in the way that they can take any type of macro-action. Future work thus can use our methods to evaluate the influence of other sets of macro-actions.
>
> Q2: We think using MAPPO would potentially lead to an unfair comparison with the value-based approaches proposed in [1], because the value-based methods do not apply any state-or-the-art extensions (e.g., prioritized replay or dueling networks) to improve performance. As this paper is the first work on macro-action-based policy gradients, we would like to investigate the properties of the proposed actor-critic methods without any fancy tricks. Both the value-based and policy gradient methods can be improved by such extensions but doing so would make comparison more difficult.
>
> Also, we would like to mention that the value-based approaches do not always have better sample-efficiency. For example, as we have shown in Fig. 4, our actor-critic approaches demonstrate better sample-efficiency and scalability over the value-based methods in the larger Warehouse domains.
>
> **Limitations**:
>
> We would like to mention that, in section 4.2, we first discussed the limitations of each proposed actor-critic framework under different domains (line 281-321), and successively, we further analyze the pros and cons of our methods by comparing with the DQN-based approaches proposed in [1]. We will add a more general discussion to the paper.

---

> > ### Comment · Reviewer_6Bsf · 2022-08-06
> > **Reviewer 6Bsf Response to Rebuttal**
> >
> > Thank you for addressing the questions and comments. I still have some reservations about the novelty and contribution of this work and would stick with my score.

---

### Meta-Review · Area_Chair_2mpb · 2022-09-10

**Recommendation:** Accept
**Confidence:** Certain

**Metareview:**

All reviewers appreciated the clarity of writing, thorough evaluation and effectiveness of the proposed method. Some reviewers had concerns about the novelty of proposed ideas, however I feel these have been satisfactorily addressed in the authors' rebuttal. Thus, I recommend acceptance.

**Award:**

No

---

### Decision · Program_Chairs · 2022-09-14

Accept